# Advancing Veterinary Oncology: Next-Generation Diagnostics for Early Cancer Detection and Clinical Implementation

**DOI:** 10.3390/ani15030389

**Published:** 2025-01-30

**Authors:** Aya Hasan Alshammari, Takuya Oshiro, Umbhorn Ungkulpasvich, Junichi Yamaguchi, Masayo Morishita, Sura Abbas Khdair, Hideyuki Hatakeyama, Takaaki Hirotsu, Eric di Luccio

**Affiliations:** 1Hirotsu Bioscience Inc., New Otani Garden Court 22F, 4-1 Kioi-cho, Chiyoda-ku, Tokyo 102-0094, Japan; alshammari@hbio.jp (A.H.A.); oshiro@hbio.jp (T.O.); umbhorn@hbio.jp (U.U.); j.yamaguchi@hbio.jp (J.Y.); m.morishita@hbio.jp (M.M.); h.hatakeyama@hbio.jp (H.H.);; 2Clinical Pharmacy Department, College of Pharmacy, Al-Mustansiriya University, Baghdad 10052, Iraq; pharm.sura.abbas@uomustansiriyah.edu.iq

**Keywords:** molecular biomarkers, One Health, One Medicine, liquid biopsy, Artificial Intelligence-enhanced imaging, molecular diagnostics, comparative oncology, companion animal cancer, precision medicine

## Abstract

Cancer is a serious problem for dogs and cats, often diagnosed when it is already advanced and more difficult to treat. This article reviews new ways to find cancer earlier in pets, so they have a better chance of successful treatment. These methods include special scanning techniques powered by computer-based image analysis, blood tests that look for cancer markers without surgery, new ways to examine DNA or other molecules from tumors, and even a simple worm-based test to detect changes linked to cancer. By catching the disease sooner, veterinarians can suggest treatments that are more targeted and less invasive, helping pets live longer and with a higher quality of life. Because cancers in animals and humans share many features, these discoveries can guide research for both species. The goal is to make advanced cancer testing available and affordable in more clinics, leading to earlier detection and better outcomes. Ultimately, this work benefits families, pets, and the broader scientific community.

## 1. Introduction

Cancer is the leading cause of mortality in companion animals, notably dogs and cats, and it poses an escalating challenge within veterinary medicine. While human oncology spans several decades of risk accumulation, companion animals face compressed lifespans, leading to annual cancer incidence rates higher than those observed in humans [1]. Among dogs, nearly half of those older than ten years are likely to develop cancer. Although systematic data for cats are less comprehensive, their cancer-related morbidity and mortality rates are similarly troubling [1].

These patterns underscore the urgent need for more effective tools to diagnose and manage cancer at its earliest, most treatable stages—a critical gap not fully addressed by conventional diagnostics, which often detect disease too late for optimal therapeutic intervention [2].

Companion animals are integral members of households worldwide, reinforcing the demand for high-quality veterinary care (Figure 1). For instance, the United States alone hosts approximately 78 million dogs and 74 million cats, while other countries also harbor substantial populations of companion animals: Brazil with an estimated 54 million dogs and China with about 53 million cats. Even nations with comparatively fewer pets, such as Japan (9 million dogs and 7 million cats), maintain high standards of veterinary care that would benefit from improved early cancer detection [3,4]. Pet-owning communities in Russia, Germany, and the United Kingdom similarly demonstrate increasing expectations for advanced veterinary oncology services, mirroring trends in human healthcare [3,4]. These global dynamics emphasize the pressing need for innovative, accessible, and regionally adaptable cancer diagnostic technologies [5].

Common canine and feline tumor profiles provide further evidence of this necessity. In dogs, skin tumors are notably frequent, representing about 49.5% of all tumors, with nearly half (48.4%) proving malignant. Mammary gland tumors in female dogs commonly exhibit a malignancy proportion of approximately 40–50%. Oral tumors, accounting for ~6% of all canine neoplasms, have a malignancy proportion of around 39%. While bone tumors are comparatively less common, when they do occur, they are malignant in roughly 90% of cases. Hematopoietic tumors, such as lymphoma, are almost invariably malignant and thus remain a persistent clinical challenge [6].

Cats present a similarly concerning scenario. Skin tumors have a prevalence of 39.6%, with about 63% showing malignant behavior. Mammary gland tumors in female cats are especially alarming, with an ~85% malignancy proportion—significantly higher than that observed in dogs [6]. Oral tumors in cats, representing approximately 3–12% of all feline tumors, are malignant in ~78% of cases. Moreover, digestive system tumors in cats have a prevalence of ~9%; many of these are intestinal lymphomas, with about 80% malignancy. Although bone tumors in cats are relatively infrequent, over 90% are malignant [6]. These data emphasize that cancer is both common and severe in companion animals, necessitating diagnostic approaches that can identify malignancies early to improve therapeutic outcomes.

Beyond clinical implications, the emotional and financial burdens of companion animal cancer are substantial. As pets increasingly assume roles analogous to human family members, owners expect compassionate care supported by state-of-the-art diagnostics. Yet, current veterinary oncology often lags behind human oncology in adopting emerging technologies [2,7].

Comparative oncology further amplifies the significance of these advances. Naturally occurring cancers in companion animals share key biological and molecular characteristics with human malignancies, providing valuable models that enhance our understanding of tumor biology, progression, and therapeutic response [8,9]. Cross-species insights have already accelerated human oncology breakthroughs, such as limb-sparing surgical techniques informed by canine osteosarcoma research and improved understanding of hormonal influences in human breast cancer through feline mammary tumor studies [2,10,11,12,13,14,15]. By adopting a “One Health” perspective—integrating veterinary and human medicine with environmental considerations—these reciprocal insights expedite progress in cancer diagnostics and therapeutics across species that support the unitary concept of “One Medicine” [16,17]. Leveraging companion animal cancer models can thus drive innovation that benefits both animal and human patients, improving global cancer research efficiency and reducing redundant efforts [9,18].

In light of these considerations, this review will (1) evaluate emerging diagnostic technologies with the potential to improve early cancer detection in companion animals, (2) examine the accessibility and adaptability of these new tools across diverse clinical settings, and (3) explore the contributions of comparative oncology in refining diagnostic precision and guiding more effective treatment strategies. Through these objectives, we aim to delineate a pathway toward advanced, accessible, and globally relevant cancer diagnostics that enhance animal welfare while informing human oncology and public health.

## 2. Innovative Diagnostic Approaches in Veterinary Oncology: Advancing Early Cancer Detection

Building on the urgent need for more effective and accessible cancer diagnostics, a range of next-generation approaches—including Artificial Intelligence-driven imaging, liquid biopsies, molecular and genetic profiling, and nematode-based screening—has emerged. These tools promise non-invasive or minimally invasive methods that enable earlier tumor detection, improved prognostication, and targeted treatment strategies [19,20]. Ensuring that these technologies remain practical, cost-effective, and broadly accessible—from urban specialty clinics to rural practices—is essential for maximizing their clinical impact.

### 2.1. Artificial Intelligence-Enhanced Imaging: Expanding the Diagnostic Lens

Artificial Intelligence (AI) is transforming veterinary oncology, especially in diagnostic imaging, by enhancing tumor detection and characterization in companion animals and addressing challenges related to accessibility and early diagnosis [7,21]. Conventional imaging techniques, such as Magnetic Resonance Imaging (MRI), Computed Tomography (CT), and ultrasound, have long been essential tools for cancer diagnosis in veterinary medicine [22]. However, the diagnostic accuracy of these methods is often influenced by the skill and expertise of individual radiologists, contributing to variability in results. This issue is particularly pronounced in veterinary settings, where differences in training and resources across practices may affect diagnostic consistency [22,23]. AI-driven tools have the potential to mitigate these challenges by delivering precise, efficient analyses that standardize diagnostics across clinics, thereby increasing access to reliable diagnostic results [24]. Moreover, as AI algorithms continue to improve with exposure to larger and more diverse datasets, their predictive power and clinical utility are expected to increase significantly. Ultimately, this convergence of advanced imaging and AI-driven interpretation is set to raise the standard of care in veterinary oncology.

The integration of AI in veterinary imaging relies on sophisticated machine learning algorithms, particularly deep learning models like convolutional neural networks (CNNs). These models are trained on large datasets of annotated medical images to identify subtle patterns and abnormalities that may be challenging for human observers to detect [25]. By automating this process, AI systems enhance diagnostic accuracy, reduce inter-observer variability, and provide reproducible results. Recent research underscores the effectiveness of AI in veterinary oncology (Table 1). For instance, a study by Aubreville et al. (2020) demonstrated that deep learning algorithms outperformed veterinary pathologists in detecting mitotically active tumor regions in canine mast cell tumors, with a correlation between predicted and ground truth mitotic counts ranging from 0.963 to 0.979 [26].

AI is also proving effective in improving diagnostic performance for imaging modalities like MRI and CT. A recent study highlighted a deep learning algorithm’s ability to classify meningeal-based and intra-axial lesions in canine brain MRI scans with over 90% accuracy, a task often considered complex for radiologists [27]. Another study found that AI-assisted MRI improved the precision of tumor volume measurements compared to manual methods, while AI-driven ultrasound reduced false-negative rates for liver tumors in dogs, showcasing its potential to mitigate human diagnostic errors and enhance clinical decision-making [7].

Integrating AI-enhanced imaging with molecular diagnostics represents a transformative approach to veterinary oncology. Molecular techniques such as Polymerase Chain Reaction (PCR) and Next-Generation Sequencing (NGS) provide detailed genetic and molecular profiles of tumors, complementing imaging data to create comprehensive diagnostic profiles. For example, AI-driven imaging can delineate tumor boundaries with remarkable precision, while molecular diagnostics identify actionable genetic mutations, enabling personalized treatment strategies. This combination reduces the need for redundant testing, optimizes cost-effectiveness, and improves treatment outcomes [28,29,30].

While research in veterinary oncology continues to expand, lessons from human medicine highlight the potential for AI in veterinary applications. A deep learning framework in human oncology achieved a balanced accuracy of 92% in differentiating molecular subtypes of lower-grade gliomas using 3D whole-brain MRI scans, demonstrating non-invasive and interpre1 advancements. Such non-invasive techniques illustrate the potential of AI to revolutionize diagnostic capabilities, particularly when extended to veterinary oncology [31]. Applying similar frameworks in veterinary medicine could enable precise tumor classification and subtype differentiation, facilitating early detection and personalized treatment for companion animals and ultimately enhancing clinical outcomes.

AI-enhanced imaging represents a groundbreaking advancement in veterinary oncology by addressing variability in diagnostic practices, enabling earlier cancer detection, and supporting precision medicine through integration with molecular diagnostics [2,24,32]. By improving accessibility, ensuring diagnostic consistency, and enabling cost-effective interventions, AI brings veterinary oncology closer to the precision of human healthcare. This technology is poised to transform cancer care for companion animals, offering a more nuanced, reliable, and individualized approach to diagnosis and treatment [32].

### 2.2. Liquid Biopsies: A New Frontier in Non-Invasive Diagnostics

Liquid biopsies introduce a transformative capability in oncology (Table 1), allowing for the detection of circulating biomarkers—such as circulating tumor DNA (ctDNA) and circulating tumor cells (CTCs)—via minimally invasive blood draws [33,61,62]. This non-invasive technology, already widely used in human oncology for precision medicine applications, holds similar promise in veterinary oncology by providing accessibility, early detection, and personalized treatment options for companion animals [34]. By integrating liquid biopsies with NGS and genomic profiling, veterinarians gain dynamic insights into tumor biology, advancing diagnostic accuracy and enabling more individualized treatment [29,34].

Liquid biopsies involve analyzing ctDNA and CTCs in bodily fluids, primarily blood. Tumor cells release fragmented DNA into the bloodstream through processes like apoptosis and necrosis [35,61]. By extracting and sequencing this cell-free DNA (cfDNA), clinicians can identify tumor-specific genetic alterations without requiring invasive tissue biopsies [35,61]. NGS technologies facilitate comprehensive genomic profiling of these cfDNA fragments, allowing for detecting mutations, copy number variations, and other genomic alterations associated with cancer [36].

One significant advantage of liquid biopsies is their ability to capture the genetic heterogeneity within tumors, including subclonal populations that traditional tissue biopsies may miss [37]. Tumors frequently exhibit genetic heterogeneity, with distinct mutations across regions that drive treatment resistance and disease progression [38]. Unlike tissue biopsies, which provide localized snapshots, liquid biopsies sample biomarkers across the circulatory system, allowing for real-time monitoring of dominant clones and emerging subclones [39]. This capability is especially valuable in veterinary oncology, where early cancer detection remains challenging, and many cancers are diagnosed at advanced stages, limiting treatment options [40].

Moreover, liquid biopsies facilitate the identification of actionable mutations in genes like EGFR, KRAS, and PIK3CA, which are implicated in both human and animal veterinary cancers [34,41]. Detecting these mutations enables veterinarians to apply targeted therapies, aligning treatments with each tumor’s unique molecular profile [40]. Such personalized approaches promise to improve companion animals’ survival and quality of life, highlighting the role of early detection and cost-effectiveness in expanding access to quality care [42].

Furthermore, genomic data from liquid biopsies support predictive modeling of tumor behavior, which is essential for proactive cancer management. Through bioinformatics, liquid biopsy data reveal metastatic potential, resistance profiles, and likely responses to various therapies, allowing veterinarians to adjust treatments dynamically as the molecular profile evolves [33,35]. In human oncology, liquid biopsies are increasingly used to monitor minimal residual disease (MRD) and potential relapse [43]; veterinary applications may soon adopt similar protocols to enhance proactive management and minimize recurrence risks [40].

However, challenges remain in establishing sensitivity and specificity benchmarks for veterinary applications. Unlike human medicine, which benefits from extensive ctDNA databases for data interpretation, veterinary oncology lacks species-specific reference databases. Large-scale initiatives akin to The Cancer Genome Atlas (TCGA) [44] could provide standardized resources for veterinary cancers, facilitating more accurate data interpretation and expanding accessibility to consistent diagnostics [45]. Refining ctDNA detection methods across species and cancer types is also essential, as ctDNA kinetics can vary widely [46].

Integrating liquid biopsy data with imaging techniques, like MRI, offers promise in veterinary oncology by combining genetic insights with detailed tumor morphology. While AI in veterinary imaging is still emerging, its potential to analyze and correlate imaging features with genetic data could enable precise treatment planning and monitoring. This approach aims to enhance diagnostic precision, enable timely interventions, and improve cost-effectiveness, ultimately benefiting patient outcomes [40,47].

In summary, integrating liquid biopsies with genomic data and advanced imaging establishes a new level of precision in veterinary oncology. It enables more individualized treatment strategies, improves patient outcomes, and underscores the importance of addressing challenges, such as species-specific ctDNA sensitivity and comprehensive database development, to fully leverage liquid biopsies in companion animal cancer care [40].

#### Proteomic, Lipidomic, and Vibrational Spectroscopy Approaches

In addition to ctDNA and CTC-based assays, proteomic and lipidomic analyses are gaining attention as novel molecular diagnostic tools for canine cancers. For instance, Gutierrez-Riquelme et al. [48] utilized a proteomic approach to identify disease-specific protein signatures in extracellular vesicles (EVs) derived from canine mammary tumor cell lines, highlighting their potential as early biomarkers of malignancy. Similarly, Enginler et al. [49] demonstrated that a lipidomics-focused electrochemical detection method, enhanced by machine learning-driven prediction, could improve sensitivity and specificity for canine mammary tumor biomarkers. By capturing additional molecular changes—beyond genetic alterations—these “omics” approaches provide a comprehensive view of tumor biology and may facilitate early detection, prognosis, and therapy selection in veterinary oncology [48,49]. As proteomic and lipidomic methods continue to mature and cost barriers decrease [50,51], integrating them alongside ctDNA-based liquid biopsies could significantly broaden the diagnostic and prognostic toolkit available to practitioners [34,40].

Moreover, vibrational spectroscopy techniques—such as infrared (IR) microspectroscopy and Raman spectroscopy—are emerging as complementary methods for real-time tissue classification, potentially enhancing pathologists’ workflow and reducing costs and turnaround times. In human-focused research, Ferguson et al. [52] demonstrated how IR microspectroscopy combined with machine learning can reliably differentiate cancerous and non-cancerous tissues under various preprocessing schemes, indicating a rapid, label-free diagnostic avenue. In the veterinary domain, Dantas et al. [53] showed that Raman spectroscopy can effectively discriminate benign from malignant canine mammary lesions, highlighting its practical utility for on-site or pilot diagnostics. Although these vibrational spectroscopy approaches are still underexplored in companion animals [52,53], they could be adapted to larger cohorts of canine and feline tumors to validate cost-effectiveness, time efficiency, and accuracy [34,40]. When integrated into a multimodal diagnostic strategy that includes proteomic, lipidomic, and ctDNA analyses [48,49], such spectroscopy-based techniques may significantly advance early detection and precision oncology in veterinary practice.

### 2.3. Nematode-Based Screening Tests Utilizing Caenorhabditis Elegans

Nematode-based screening utilizing *Caenorhabditis elegans* (*C. elegans*) offers an innovative and promising approach to early cancer detection in veterinary oncology (Table 1), with particular emphasis on accessibility and affordability [54,55]. Unlike molecular or imaging techniques that require specialized equipment and substantial technical expertise, this method leverages the olfactory capabilities of *C. elegans* to detect cancer-associated volatile organic compounds (VOCs) present in bodily fluids, such as urine [56,57]. Preliminary studies suggest that *C. elegans* can distinguish between samples from healthy individuals and those with cancer, demonstrating significant potential as an initial screening tool [55,56,58]. However, while this approach provides an early indication of potential cancer, it is not intended to replace confirmatory diagnostic methods.

Research indicates that *C. elegans* may detect cancer-associated VOCs with notable sensitivity and specificity, supporting its promise as a non-invasive screening method [58,59,60]. This capability is particularly relevant in veterinary settings, where early detection is frequently hindered by subtle or delayed symptoms, making timely intervention challenging [63]. The affordability and simplicity of *C. elegans* screening further highlight its potential as a practical solution for improving early screening efforts in veterinary medicine, particularly for companion animals [55].

The application of *C. elegans*-based screening has shown encouraging results in both experimental and translational studies. In human trials, *C. elegans* has demonstrated its ability to identify early-stage cancers, including pancreatic and gastric cancers, often before clinical symptoms emerge [56,64,65,66]. This early detection capability could hold significant implications for veterinary oncology, where early-stage cancers in companion animals, such as dogs and cats, often go unnoticed due to a lack of apparent clinical signs [40]. By integrating *C. elegans* screening into routine veterinary care, practitioners could flag potential cancer cases for further evaluation—such as imaging or biopsy—facilitating earlier intervention and improving outcomes.

Beyond its screening potential, *C. elegans* tests offer exceptional cost-effectiveness and simplicity, making them an accessible tool even for veterinary clinics with limited resources. Compared to the substantial costs associated with advanced imaging modalities or molecular diagnostics, this method provides a budget-friendly, non-invasive alternative for pet owners. Its affordability and scalability make it particularly suitable for high-risk or aging pets and for use in regular health assessments. The adaptability of this approach supports its broad implementation in various clinical settings, thereby expanding access to quality veterinary oncology care and enhancing early detection efforts [56].

In summary, *C. elegans*-based screening represents a transformative advancement in non-invasive cancer detection for veterinary applications. While it serves primarily as a preliminary screening tool, its high sensitivity, simplicity, and cost-effectiveness make it a valuable addition to the veterinary screening landscape. This method complements existing technologies by enabling accessible early detection, which can significantly improve treatment options for companion animals [56]. As ongoing research continues to validate its efficacy and expand its applications, *C. elegans* screening has the potential to become an integral part of proactive cancer care in veterinary medicine, bridging innovative, affordable methods with confirmatory diagnostics to facilitate timely and effective interventions.

## 3. Comparative Oncology: A Shared Path to Advancing Cancer Care for Humans and Pets

Comparative oncology is a critical interdisciplinary field investigating naturally occurring cancers in companion animals, particularly dogs and cats, to provide translational insights into human oncology. By examining shared biological, environmental, and lifestyle factors, comparative oncology facilitates a bidirectional exchange of knowledge that accelerates advancements in cancer diagnostics, treatment protocols, and early detection strategies. This approach not only promotes cost-effective and accessible solutions but also enriches our understanding of cancer biology across species [9].

One of the most illustrative examples of comparative oncology is osteosarcoma research. Dogs naturally develop osteosarcoma at a rate 10 times higher than humans, with similar histological features, genetic mutations, and metastatic behavior. As a result, they serve as invaluable models for studying disease progression and therapeutic efficacy [67].

For instance, before human applications, HER2-targeted therapies, such as HER2-specific monoclonal antibodies, were trialed in canine osteosarcoma models. These studies informed the development of HER2-targeted immunotherapies like trastuzumab, widely used in human HER2-positive cancers [68,69]. Furthermore, research on metronomic chemotherapy in dogs—administering low-dose chemotherapeutics to inhibit angiogenesis—has directly shaped human treatment regimens for pediatric osteosarcoma, improving survival rates while minimizing toxicity [70].

Canine lymphoma research further underscores the contributions of comparative oncology. This disease mirrors human non-Hodgkin’s lymphoma in terms of biological behavior and treatment response, enabling significant advancements in therapy. The CHOP chemotherapy protocol—comprising cyclophosphamide, doxorubicin, vincristine, and prednisone—was optimized through studies in canine lymphoma and has since become a cornerstone of human lymphoma treatment. These shared advancements have substantially improved survival rates and quality of life in both species, exemplifying the mutual benefits of cross-disciplinary research [14,71,72].

Certain dog breeds, such as Scottish Terriers, have a predisposition to transitional cell carcinoma (TCC), analogous to high-grade muscle-invasive bladder cancer in humans [73]. Studies on the genetic basis of canine TCC, particularly the BRAF V600E mutation, have directly influenced human precision oncology. The development of vemurafenib, a BRAF inhibitor, was guided by these canine studies and has become a cornerstone therapy for BRAF-mutant cancers, including bladder and melanoma in humans [74]. Recent research has also identified common environmental risk factors in both species, such as exposure to herbicides and carcinogenic byproducts. These findings have prompted public health interventions to reduce exposure and improve preventive strategies across species [74].

Mammary tumors in dogs exhibit a high degree of genetic, hormonal, and molecular similarity to human breast cancer, particularly the triple-negative subtype. Studies on canine mammary tumors have led to breakthroughs in targeted therapies, such as PARP inhibitors for BRCA1/BRCA2 mutations. These findings have directly informed human trials, resulting in the FDA approval of olaparib for BRCA-mutant breast cancers [75,76]. Additionally, canine trials of anti-angiogenic therapies have refined the use of drugs like bevacizumab in human breast cancer, demonstrating improved efficacy in reducing tumor vascularization and progression [77].

Human advancements also benefit veterinary oncology. Next-generation sequencing (NGS) technologies, initially developed for human cancer research, have been adapted for tumor profiling in companion animals, enabling personalized treatment strategies. For example, NGS applications in canine glioblastoma have guided precision therapies that mirror human glioblastoma management, demonstrating the reciprocal flow of knowledge [78,79]. Furthermore, collaborative studies in cancer immunotherapy, such as checkpoint inhibitors targeting PD-1/PD-L1 pathways, have shown remarkable promise in both dogs and humans. Canine trials have accelerated the development of these therapies for human use while simultaneously improving outcomes for veterinary patients [80].

Comparative oncology exemplifies the power of collaborative, cross-disciplinary research. It bridges the gap between veterinary and human medicine by fostering innovation in early detection, precision therapies, and cost-effective treatments [40,81]. This field has driven the development of therapies like HER2-targeted immunotherapy, PI3K inhibitors, BRAF inhibitors, and PARP inhibitors, underscoring its transformative impact on cancer care. As a cornerstone of the “One Health” philosophy, comparative oncology highlights the interconnected nature of human and animal health. By leveraging naturally occurring cancer models, this field continues to propel advancements that benefit humans and companion animals, offering a unified framework for improving cancer outcomes across species [16,17].

## 4. Challenges in Implementing Cutting-Edge Diagnostics in Veterinary Practice

While advanced diagnostic technologies promise substantial improvements in early cancer detection and personalized pet care, several practical challenges impede their widespread implementation. High costs, limited access to specialized equipment, and a need for skilled training hinder the reach and effectiveness of advanced diagnostics in veterinary oncology. Addressing these barriers requires strategic and innovative solutions to make diagnostics—such as AI-enhanced imaging, liquid biopsies, and molecular diagnostics—more accessible, cost-effective, and seamlessly integrated into everyday veterinary care [82].

### 4.1. Financial Barriers and Accessibility

The considerable costs associated with advanced diagnostic technologies—including MRI, CT scans, liquid biopsies, and molecular testing—present a major obstacle to their widespread adoption, particularly in general veterinary practices and rural clinics [83]. These financial constraints often confine such tools to urban or well-funded specialty centers, limiting their availability to pet owners in underserved regions [50]. As a result, animals in economically disadvantaged areas may face delayed diagnoses, which can lead to more advanced diseases at the time of detection and reduced treatment options [50,84]. Even when AI-driven imaging proves highly effective at identifying early-stage cancers, the initial investment required to implement and maintain these systems often remains out of reach for many smaller practices [83].

To address these financial barriers, several innovative strategies can be employed to enhance the accessibility of advanced diagnostics. Pet insurance plans that cover preventive screenings and diagnostic procedures can alleviate the immediate financial burden on owners, encouraging earlier and more frequent evaluations. Comprehensive pet insurance policies often include coverage for diagnostic tests, such as blood tests, urinalyses, X-rays, and ultrasounds. The availability of such coverage enables pet owners to pursue necessary diagnostic procedures without undue financial strain [84].

At the same time, community-led assistance programs and partnerships with nonprofit organizations may provide crucial support for low-income clients [50]. Emerging cost-effective screening tools, such as *C. elegans*-based screening, represent another promising avenue for expanding accessibility. Such technologies could offer a more affordable alternative to traditional methods, making early detection feasible for a broader demographic [56,59,85].

As diagnostic technologies mature, increased market competition and scaled-up production can lead to cost reductions. In human medicine, the widespread adoption of next-generation sequencing has significantly decreased sequencing costs, making these technologies more accessible over time. A similar trend in veterinary diagnostics could further alleviate financial barriers [51].

Enhancing the affordability of advanced diagnostic tools is directly linked to improved clinical outcomes and increased client satisfaction. Accessible diagnostics facilitate early detection, leading to more effective and potentially less expensive treatments. This creates a positive feedback loop, where improved outcomes boost client willingness to invest in diagnostics, promoting a proactive approach to animal healthcare [86].

By exploring and implementing these financial strategies, the field can move closer to a future where high-quality diagnostic tools are available to a wider range of veterinary practices. This shift would promote earlier detection and intervention and improve overall patient outcomes, ensuring that the benefits of cutting-edge oncology tools are accessible to all companion animals and their caregivers [50,83].

### 4.2. Infrastructure Challenges and Technological Adaptation

Infrastructure limitations represent a significant barrier to integrating advanced diagnostic technologies in veterinary oncology, particularly in resource-limited settings such as smaller clinics and rural areas. Sophisticated tools, including MRI, CT scanners, and AI-enhanced imaging systems, demand specialized infrastructure, uninterrupted power supplies, and trained personnel—resources often unavailable outside well-funded specialty centers. These limitations exacerbate disparities in diagnostic care, leaving underserved regions with delayed or inaccurate cancer diagnoses, which negatively impacts treatment outcomes [50,83].

To address these challenges, innovative and infrastructure-adaptable solutions are essential. Portable diagnostic tools, such as handheld ultrasound devices and compact point-of-care molecular analyzers, have emerged as transformative technologies. For example, wireless probe-based ultrasound systems integrated with telemedicine functionalities enable veterinarians to conduct real-time diagnostics without extensive laboratory setups. Similarly, portable PCR devices now allow on-site genetic analysis, bypassing the need for centralized laboratories. Mobile diagnostic units equipped with imaging modalities and screening technologies extend these capabilities to geographically isolated communities, effectively bridging the gap in accessibility [87,88].

Cloud-based platforms and telemedicine further alleviate infrastructural barriers by enabling remote diagnostic interpretation and expert consultations. Diagnostic data, including imaging scans and molecular test results, can be securely transmitted to specialists, providing high-quality analysis in real time. Telerobotic ultrasound technology has demonstrated efficacy in enabling remote ultrasound guidance, significantly enhancing diagnostic precision in underserved settings. These technologies not only democratize access to specialized expertise but also streamline diagnostic workflows, reducing delays in care delivery [89,90,91].

Collaborative efforts between technology developers, veterinary institutions, and policymakers are essential for scaling these innovations sustainably. Public–private partnerships and targeted funding initiatives can support the deployment, maintenance, and training of advanced diagnostic tools. Legislative efforts, such as the Rural Veterinary Workforce Act, aim to expand access to veterinary care in underserved areas, underscoring the importance of systemic approaches to addressing these challenges [50,92,93].

In summary, overcoming infrastructural barriers through portable diagnostics, mobile units, and telemedicine is essential for democratizing veterinary oncology. These innovations, coupled with strategies to reduce financial constraints, will enable earlier cancer detection, enhance diagnostic accuracy, and expand access to life-saving technologies for companion animals across diverse settings [92].

### 4.3. Advancing Diagnostic Proficiency: Education and Training for Emerging Technologies

The successful integration of advanced diagnostic tools into veterinary oncology hinges on addressing significant skill gaps among practitioners. Technologies such as AI-enhanced imaging, liquid biopsies, and molecular diagnostics necessitate specialized proficiencies in bioinformatics, data interpretation, and molecular analysis—not traditionally emphasized in veterinary training programs. For example, interpreting AI-generated imaging outputs requires an understanding of machine learning algorithms and the ability to critically evaluate algorithmic conclusions. Similarly, analyzing genomic data from liquid biopsies involves navigating complex datasets, necessitating expertise in genomics and molecular biology [40,94].

Targeted educational initiatives are critical to address these challenges. Certification programs, workshops, and online courses can equip veterinarians with the skills necessary to utilize emerging diagnostic technologies effectively. For instance, the American College of Veterinary Internal Medicine (ACVIM) offers educational modules focusing on the practical application of personalized and genomic diagnostics in veterinary oncology [95]. These programs aim to bridge the gap between theoretical knowledge and clinical implementation, ensuring practitioners can deliver precision diagnostics in routine practice [96].

Collaborations between veterinary schools, technology developers, and industry stakeholders can further enhance training opportunities [97]. Comparative oncology research, which bridges veterinary and human oncology, offers a unique opportunity for veterinarians to collaborate with oncologists, geneticists, and computational biologists. For example, the Mari Lowe Center for Comparative Oncology Research at the University of Pennsylvania fosters cross-disciplinary research and training to develop diagnostics and treatments that benefit human and animal patients [98]. These initiatives underscore the value of translational research in enhancing diagnostic competency and advancing veterinary oncology [99].

Adopting e-learning and tele-training platforms is vital for democratizing access to education, particularly in remote or resource-constrained settings. Webinars, virtual simulations, and teleconsultations offer scalable and flexible training options, enabling veterinarians to stay abreast of technological advancements irrespective of geographical limitations. For example, the World Small Animal Veterinary Association (WSAVA) provides online oncology resources and virtual workshops to make specialized training more accessible worldwide [100].

Systematic evaluation and continuous improvement ensure that training programs remain effective. Regular assessments of knowledge acquisition, refresher courses, and “train-the-trainer” models help maintain a skilled workforce. This approach fosters lifelong learning, ensuring veterinarians can adapt to advancements in diagnostic technologies [101].

Enhanced diagnostic proficiency directly improves patient outcomes and builds client trust. Veterinarians skilled in advanced diagnostics achieve earlier detection and more accurate diagnoses, leading to better treatment outcomes. Communicating the benefits of these technologies further encourages client adoption, raising the standard of care in veterinary oncology [102].

Addressing educational needs is essential for successfully integrating emerging diagnostic tools into veterinary oncology. Through targeted education, interdisciplinary collaboration, and continuous assessment, veterinarians can leverage these advancements to improve patient care, diagnostic accuracy, and client satisfaction, driving progress in the field [86].

## 5. Conclusions

Next-generation diagnostics—including AI-enhanced imaging [7,21], liquid biopsies [29,40], molecular assays [28,103], and nematode-based screening [54,55,56,57,58,59]—demonstrate enormous potential for improving the early detection of cancer in companion animals. To fully realize this potential, cost, infrastructure, and training barriers must be addressed [83,87], ensuring more equitable access to these innovations in diverse veterinary contexts. Equally important is the role of comparative oncology: naturally occurring tumors in pets continue to provide translational insights that inform both veterinary and human cancer research [10,45]. By adopting these advanced diagnostic tools within a “One Health, One Medicine” framework [16,17,19,81], the field can elevate the standard of care for companion animals while fostering progress in global cancer prevention and therapy.

## Figures and Tables

**Figure 1 animals-15-00389-f001:**
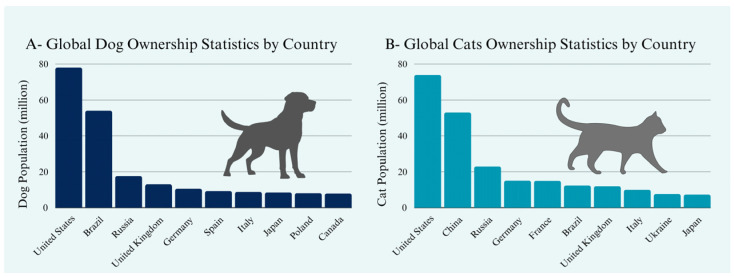
(**A**) Dog populations in selected countries based on estimated ownership data. (**B**) Cat populations in selected countries based on estimated ownership data. Data sourced from World Population Review (2024) [3,4].

**Table 1 animals-15-00389-t001:** Comparative Overview of Next-Generation Diagnostic Modalities in Veterinary Oncology.

Approach	Mechanism/Technology	Advantages	Challenges	Ref
AI-Enhanced Imaging	Deep learning algorithms (e.g., CNNs) trained on annotated imaging datasets (MRI, CT, ultrasound)	- Increased diagnostic accuracy and consistency - Reduced inter-observer variability - Earlier tumor detection - Integration with molecular data for comprehensive profiling	- Requires high-quality, annotated datasets - Variable implementation costs - Ongoing validation for diverse tumor types	[8,21,26,27,28,29,30,31,32]
Liquid Biopsies (ctDNA, CTCs)	Non-invasive analysis of cell-free tumor DNA or circulating tumor cells from blood samples, often integrated with NGS	- Minimally invasive - Captures tumor heterogeneity - Facilitates personalized treatment strategies - Potential for real-time monitoring and early detection	- Limited species-specific reference databases - Variable ctDNA kinetics across cancer types - Need for standardized sensitivity/specificity benchmarks	[29,33,34,35,36,37,38,39,40,41,42,43,44,45,46,47,48,49]
Nematode-Based Screening (*C. elegans*)	Detection of cancer-associated volatile organic compounds (VOCs) in bodily fluids (e.g., urine) using olfactory capabilities of *C. elegans*	- Low-cost, simple, and scalable - Non-invasive - Potentially useful as an early, accessible screening tool, especially in resource-limited settings	- Primarily a preliminary screening measure	[50,51,52,53,54,55,56,57,58,59,60]

## Data Availability

No new data were created or analyzed in this study.

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
