# Peer review of "Advancing Veterinary Oncology: Next-Generation Diagnostics for Early Cancer Detection and Clinical Implementation"

_animals, 2025, doi:10.3390/ani15030389_

Round 1

Reviewer 1 Report

Comments and Suggestions for Authors

This is a relatively comprehensive and easy to be understood by the majority of veterinarians. The new techniques for diagnosing tumor diseases in dogs and cats, including artificial intelligence, are described comprehensively. To provide reference materials for fellow veterinarians. The author also explained the relationship between one of the authors and the content in the last conflict of interest, and the data presented in the article did not have a significant impact on the conclusion.

Author Response

Reviewer 1

Comments 1

“This is a relatively comprehensive and easy to be understood by the majority of veterinarians… The author also explained the relationship between one of the authors and the content in the last conflict of interest, and the data presented in the article did not have a significant impact on the conclusion.”

Response 1

Thank you for your positive feedback. We are pleased you find our review accessible for veterinarians and that our conflict-of-interest statement is clear and transparent. Since no specific changes were requested, no revisions were necessary for this point. However, we confirm that the disclosed conflict of interest does not impact our data analysis or conclusions.

Reviewer 2 Report

Comments and Suggestions for Authors

Dear Authors, the review is fine and needs very minimal improvements as follow.

Line 107 - after the word species insert the sentence "that support the unitary concept "One Medicine";

Line 158 - Please, remove the space separating the word "pr ovide";

Line 210 - Please, enter the word "animal" before veterinary ...;

Line 519 - Please, enter the concept "One Medicine" between "One Heath" ... framework.

Author Response

Note: The manuscript was reformatted, causing changes in line references. We indicate both the original line number (from the previous version) and the new line number in our revised manuscript (manuscript.v4.docx). 

Reviewer 2

Comments 1

“Line 107 - after the word ‘species’ insert the sentence ‘that support the unitary concept One Medicine’;
Line 158 - remove the space separating ‘pr ovide’;
Line 210 - enter the word ‘animal’ before ‘veterinary’;
Line 519 - enter the concept ‘One Medicine’ between ‘One Health’… framework.”

Response 1

We appreciate these specific editorial suggestions. Below is a breakdown of each change:

  1. Original line 107 → new line 100
    • Change: Inserted the phrase “…that support the unitary concept ‘One Medicine’” immediately after “species.”
    • Location: Approx. Page 3, Paragraph 2, Lines 99–101 in the revised manuscript.
    • Updated text in red:

“…across species that support the unitary concept ‘One Medicine’ [16,17].”

  1. Original line 158 → new line 160
    • Change: Removed the spacing error so it now reads “provide” instead of “pr ovide.”
    • Location: Approx. Page 4, Paragraph 4, Line 160 in the revised manuscript.
    • Updated text in red:

“…Next-Generation Sequencing (NGS) provide detailed genetic and…”

  1. Original line 210 → new line 212
    • Change: Added the word “animal” before “veterinary,” so it reads “animal veterinary.”
    • Location: Approx. Page 5, Paragraph 6, Line 212 in the revised manuscript.
    • Updated text in red:

“…understanding, emphasizing animal veterinary applications…”

  1. Original line 519 → new line 530
    • Change: Inserted “One Medicine” alongside “One Health” in the conclusion.
    • Location: Approx. Page 12, Paragraph 1, Line 530 in the revised manuscript.
    • Updated text in red:

“…By adopting these advanced diagnostic tools within a ‘One Health, One Medicine framework…”

These edits are highlighted in red in the revised manuscript. Thank you for your attention to these details.

Reviewer 3 Report

Comments and Suggestions for Authors

General comment: This review paper highlights the developments in diagnostic technologies and their application in veterinary oncology, enhancing early detection, accessibility, and precision in cancer care. 

The paper fits the scope of the journal, and the writing style is adequate. 

Title: The title is short, concise, and adequate.

Abstract and keywords: The abstract is complete. The keywords should be different from those used in the tile. 

Introduction: The authors provide a good and complete overview of the thematic. The aim of this work is also clearly stated in this section.

Please provide the abbreviation in the first use of “Artificial Intelligence”. (line 121)

Please review the following sentence: “A study by dvm360 highlighted a deep learning algorithm's…” (line 149)

Please review the following sentence: “…Next-Generation Sequencing (NGS) pr ovide…” (line 158)

The advantages and disadvantages of all methodologies were clearly presented in Table 1 and discusses in the manuscript. 

The conclusions of the manuscript are adequate.

The references are adequate. 

Author Response

Note: The manuscript was reformatted, causing changes in line references. We indicate both the original line number (from the previous version) and the new line number in our revised manuscript (manuscript.v4.docx). 

Reviewer 3

Comments 1

“Please provide the abbreviation in the first use of ‘Artificial Intelligence’ (line 121).
Please review the following sentence: ‘A study by dvm360 highlighted a deep learning algorithm’s…’ (line 149).
Please review the following sentence: ‘…Next-Generation Sequencing (NGS) pr ovide…’ (line 158).
The advantages and disadvantages of all methodologies were clearly presented.”

Response 1

Below is how we addressed each concern:

  1. Original line 121 → new line 122
    • Change: Defined “Artificial Intelligence (AI)” at its first mention.
    • Location: Approx. Page 3, Paragraph 1, Line 122 in the revised manuscript.
    • Updated text in red:

“…Artificial Intelligence (AI) is transforming veterinary oncology…”

  1. Original line 149 → new line 151
    • Change: Replaced the dvm360 reference with the original, peer-reviewed source (Banzato et al. 2018), Reference [27],
    • Location: Approx. Page 3, Paragraph 3, Line 151 in the revised manuscript.
    • Updated text in red:

A recent study highlighted a deep learning algorithm’s ability to classify meningeal-based…”

  1. Original line 158 → new line 160
    • Change: (Overlap with Reviewer 2) Corrected “pr ovide” to “provide.”
    • Location: Page 4, Paragraph 4, Line 160.
    • Updated text in red:

“…provide detailed genetic and molecular profiles…”

Additionally, we updated our Keywords to avoid duplication with the manuscript title. See Page 1 of the revised PDF or Word file for the final keyword list in red.

updated key words:

Keywords: Molecular Biomarkers; One Health, One Medicine; liquid biopsy; Artificial Intelligence-enhanced imaging; molecular diagnostics; comparative oncology; companion animal cancer; precision medicine

Reviewer 4 Report

Comments and Suggestions for Authors

Dear authors,

Thank you for the very interesting review on veterinary oncology diagnostics. I really enjoyed reading it, as it was well-written and well-structured. However, there are redundant parts that should be more concise. I have a few comments that might improve the quality even further that the authors need to correct or might consider adding.

The introduction is good; however, it is too long, as most of the aspects are discussed in what follows. I would like to keep introducing statistics of ownership and epidemiology, but leave parts like: “The integration of AI-driven imaging analysis, liquid biopsies, molecular and genetic diagnostics, and novel approaches—such as nematode-based screening—could transform the field. These next-generation diagnostics promise non-invasive or minimally invasive methods that enable earlier tumor detection, improved prognostication, and tailored treatment strategies [9,10]. Ensuring these tools are practical, cost-effective, and broadly accessible—from urban specialist centers to rural clinics—is essential to maximizing their impact. Over time, such innovations may help normalize advanced cancer care as a standard expectation, raising the bar for the veterinary profession as a whole. “ to latter part.

Line 149: “A study by dvm360 highlighted a deep” Please correct the reference

Lines 320-325 there is an issue with references

Line 489: The conclusions part is too wordy; it should summarize the article, not provide new insights and new references. Parts of it repeat already discussed aspects, e.g., “However, the equitable implementation of these technologies requires addressing critical barriers to accessibility, affordability, and infrastructure”, “AI-enhanced imaging provides unprecedented precision, setting a benchmark for early cancer diagnosis applicable in both veterinary and human oncology [79]. “

Please shorten the conclusions significantly.

Line 506: “Molecular diagnostics, with their ability to identify specific biomarkers, directly support targeted therapies reshaping veterinary and human cancer care [101]. This is a very good remark but there are aspects related to biomarkers that authors did not report. It would be very good to indicate the paths that human medicine is already exploring, such as proteomic, lipidomic approaches in looking for neoplastic markers. There are few works that can results in short paragraph introducing the readers with those novel diagnostic methods. As an examples:

Gutierrez-Riquelme T, Karkossa I, Schubert K, Liebscher G, Packeiser EM, Nolte I, von Bergen M, Murua Escobar H, Aguilera-Rojas M, Einspanier R, Stein T. Proteomic analysis of extracellular vesicles derived from canine mammary tumour cell lines identifies protein signatures specific for disease state. BMC Vet Res. 2024 Oct 26;20(1):488. doi: 10.1186/s12917-024-04331-1. PMID: 39462388; PMCID: PMC11515202.

Enginler SÖ, Küçükdeniz T, Dal GE, Yıldırım F, Cilasun GE, Alkan FÜ, Gürgen HÖ, TaÅŸaltın N, Sabuncu A, Yılmaz M, KarakuÅŸ S. Enhancing electrochemical detection through machine learning-driven prediction for canine mammary tumor biomarker with green silver nanoparticles. Anal Bioanal Chem. 2024 Sep;416(23):5071-5088. doi: 10.1007/s00216-024-05444-0. Epub 2024 Jul 20. PMID: 39031228; PMCID: PMC11377509.

I also feel a little omitted part of diagnostics, which can also be enhanced and what can improve work of pathologists. For example vibrational spectroscopy imaging coupled with machine learning can improve the detection and at the same time be more cost and time effective. As you wrote: “veterinary applications may soon adopt similar protocols to enhance proactive management and minimize recurrence risks”

The principles from human medicine can be found here:

Ferguson D, Henderson A, McInnes EF, Lind R, Wildenhain J, Gardner P. Infrared micro-spectroscopy coupled with multivariate and machine learning techniques for cancer classification in tissue: a comparison of classification method, performance, and pre-processing technique. Analyst. 2022 Aug 8;147(16):3709-3722. doi: 10.1039/d2an00775d. PMID: 35852144.

There are not so many papers in the field, as an example:

Dantas D, Soares L, Novais S, Vilarinho R, Moreira JA, Silva S, Frazão O, Oliveira T, Leal N, Faísca P, Reis J. Discrimination of Benign and Malignant Lesions in Canine Mammary Tissue Samples Using Raman Spectroscopy: A Pilot Study. Animals (Basel). 2020 Sep 14;10(9):1652. doi: 10.3390/ani10091652. PMID: 32937987; PMCID: PMC7552658.

It would improve the quality of your work after adding.  

Author Response

Note: The manuscript was reformatted, causing changes in line references. We indicate both the original line number (from the previous version) and the new line number in our revised manuscript (manuscript.v4.docx). 

Reviewer 4

Comments 1

“The introduction is too long, as most of the aspects are discussed in what follows… leave parts like ‘The integration of AI-driven imaging analysis…’ to the latter part.
Line 149: ‘A study by dvm360…’ correct the reference.
Lines 320–325: an issue with references.
Line 489: The conclusion is too wordy; it should summarize the article.
Line 506: … mention proteomic, lipidomic approaches, vibrational spectroscopy.”

Response 1

Thank you for these comprehensive suggestions. We have made the following revisions:

  1. Introduction Streamlining
    • Change: We shortened the introduction, moving AI, liquid biopsy, and nematode-based screening details to Section 2.
    • Location: The new Section 2 content is at lines 111–120 (approx. Page 3, Paragraph 1).
    • Updated text in red:
  1. Innovative Diagnostic Approaches in Veterinary Oncology: Advancing Early Cancer Detection

Building on the urgent need for more effective and accessible cancer diagnostics, a range of next-generation approaches—including Artificial Intelligence-driven imaging, liquid biopsies, molecular and genetic profiling, and nematode-based screening—has emerged. These tools promise non-invasive or minimally invasive methods that enable earlier tumor detection, improved prognostication, and targeted treatment strategies [19,20]. Ensuring that these technologies remain practical, cost-effective, and broadly accessible—from urban specialty clinics to rural practices— is essential for maximizing their clinical impact.

  1. Line 149 → new line 151
    • Change: As noted, replaced the reference to dvm360 with the Banzato et al. (2018) study.
    • Location: Page 3, Paragraph 3, Line 151.
  2. Lines 320–325 → new lines 351–356
    • Change: Corrected reference numbering in that section.
    • Location: Approx. Page 8, Paragraph 4, Lines 351–356.
  3. Line 489 → new line 522
    • Change: Shortened the conclusion to summarize key points without introducing new references.
    • Location: Page 12, Paragraph 1, Lines 522–531.
    • Updated text in red:

      Conclusion:

      Next-generation diagnostics—including AI-enhanced imaging [7,21], liquid biopsies [29,42] , molecular assays [28,103], and nematode-based screening [56–61]—demonstrate enormous potential for improving the early detection of cancer in companion animals. To fully realize this potential, cost, infrastructure, and training barriers must be addressed [83,87], ensuring more equitable access to these innovations in diverse veterinary contexts. Equally important is the role of comparative oncology: naturally occurring tumors in pets continue to provide translational insights that inform both veterinary and human cancer research [10,47]. By adopting these advanced diagnostic tools within a “One Health, One Medicine” framework [16,17,19,81], the field can elevate the standard of care for companion animals while fostering progress in global cancer prevention and therapy.

  4. Line 506 & Biomarker Approaches
    • Change: Added a new subsection on proteomic, lipidomic, and vibrational spectroscopy approaches (Gutierrez-Riquelme et al., Enginler et al., Ferguson et al., Dantas et al.).
    • Location: Section 2.2.1, lines 245–274 (approx. Pages 6–7).
    • Updated text in red:

2.2.1 Proteomic, Lipidomic, and Vibrational Spectroscopy Approaches

In addition to ctDNA and CTC-based assays, proteomic and lipidomic analyses are gaining attention as novel molecular diagnostic tools for canine cancers. For instance, Gutierrez-Riquelme et al. [50] utilized a proteomic approach to identify disease-specific protein signatures in extracellular vesicles (EVs) derived from canine mammary tumor cell lines, highlighting their potential as early biomarkers of malignancy. Similarly, Enginler et al. [51] demonstrated that a lipidomics-focused electrochemical detection method, enhanced by machine learning-driven prediction, could improve sensitivity and specificity for canine mammary tumor biomarkers. By capturing additional molecular changes—beyond genetic alterations—these “omics” approaches provide a comprehensive view of tumor biology and may facilitate early detection, prognosis, and therapy selection in veterinary oncology [50,51]. As proteomic and lipidomic methods continue to mature and cost barriers decrease [52,53] , integrating them alongside ctDNA-based liquid biopsies could significantly broaden the diagnostic and prognostic toolkit available to practitioners [36,42].

Moreover, vibrational spectroscopy techniques—such as infrared (IR) microspectroscopy and Raman spectroscopy—are emerging as complementary methods for real-time tissue classification, potentially enhancing pathologists’ workflow and reducing costs and turnaround times. In human-focused research, Ferguson et al. [54].  demonstrated how IR microspectroscopy combined with machine learning can reliably differentiate cancerous and non-cancerous tissues under various preprocessing schemes, indicating a rapid, label-free diagnostic avenue. In the veterinary domain, Dantas et al. [55] showed that Raman spectroscopy can effectively discriminate benign from malignant canine mammary lesions, highlighting its practical utility for on-site or pilot diagnostics. Although these vibrational spectroscopy approaches are still underexplored in companion animals [54,55], they could be adapted to larger cohorts of canine and feline tumors to validate cost-effectiveness, time efficiency, and accuracy [36,42]. When integrated into a multimodal diagnostic strategy that includes proteomic, lipidomic, and ctDNA analyses [50,51], such spectroscopy-based techniques may significantly advance early detection and precision oncology in veterinary practice

These modifications are clearly marked in red in the revised Word document.

Round 2

Reviewer 4 Report

Comments and Suggestions for Authors

Dear authors,

Thank you for sending the revised manuscript. I truly appreciate your work and I think this a very good manuscript I can recommend.